# Clinically relevant atovaquone-resistant human malaria parasites fail to transmit by mosquito

Victoria A. Balta [1,2], Deborah Stiffler[1,2], Abeer Sayeed[1,2], Abhai K. Tripathi [1,2], Rubayet Elahi[1,2], Godfree Mlambo[1,2], Rahul P. Bakshi[2,3], Amanda G. Dziedzic[1], Anne E. Jedlicka [1], Elizabeth Nenortas[3], Keyla Romero-Rodriguez[3], Matthew A. Canonizado[3], Alexis Mann[1,2], Andrew Owen [4], David J. Sullivan [1,2], Sean T. Prigge [1,2], Photini Sinnis [1,2] & Theresa A. Shapiro [1,2,3] ✉

Long-acting injectable medications, such as atovaquone, offer the prospect of a "chemical vaccine" for malaria, combining drug efficacy with vaccine durability. However, selection and transmission of drug-resistant parasites is of concern. Laboratory studies have indicated that atovaquone resistance disadvantages parasites in mosquitoes, but lack of data on clinically relevant *Plasmodium falciparum* has hampered integration of these variable findings into drug development decisions. Here we generate atovaquone-resistant parasites that differ from wild type parent by only a Y268S mutation in cytochrome *b*, a modification associated with atovaquone treatment failure in humans. Relative to wild type, Y268S parasites evidence multiple defects, most marked in their development in mosquitoes, whether from Southeast Asia (*Anopheles stephensi*) or Africa (*An. gambiae)*. Growth of asexual Y268S *P. falciparum* in human red cells is impaired, but parasite loss in the mosquito is progressive, from reduced gametocyte exflagellation, to smaller number and size of oocysts, and finally to absence of sporozoites. The Y268S mutant fails to transmit from mosquitoes to mice engrafted with human liver cells and erythrocytes. The severe-to-lethal fitness cost of clinically relevant atovaquone resistance to *P. falciparum* in the mosquito substantially lessens the likelihood of its transmission in the field.

Despite concerted efforts, that over several decades successfully cut by half the world's malaria burden, in recent years progress has stalled and appears to be reversing. Estimates in 2021 were of 247 million infections and 619,000 deaths, attesting to malaria's ongoing threat to public health and productivity[1]. Of greatest concern are infections with *Plasmodium falciparum*, which comprise 90% of cases and may be lethal, though geographically widespread *P. vivax* also causes significant morbidity. Bednets, vector reduction, various drug regimens, and, most recently, the *P. falciparum* RTS,S vaccine, are means of control, but new strategies are clearly needed. Among the candidates are long-acting injectable antimalarials[2], which for chemoprophylaxis have aptly been termed "chemical vaccines"[3].

Although long-acting injectable drugs are mainstays in contraceptive and antipsychotic therapy, new advances in drug formulation

[1]W. Harry Feinstone Department of Molecular Microbiology and Immunology, The Johns Hopkins Bloomberg School of Public Health, Baltimore, MD 21205, USA. [2]The Johns Hopkins Malaria Research Institute, Baltimore, MD 21205, USA. [3]Division of Clinical Pharmacology, Departments of Medicine and of Pharmacology and Molecular Sciences, The Johns Hopkins University, Baltimore, MD 21205-2186, USA. [4]Centre of Excellence in Long-acting Therapeutics (CELT), Department of Pharmacology and Therapeutics, University of Liverpool, Liverpool L69 3BX, UK. ✉e-mail: tshapiro@jhmi.edu

have brought this modality into use against infectious diseases[4–6]. Every-two-month injection of viral integrase inhibitor cabotegravir was recently approved as monoprophylaxis against sexually acquired HIV infection[7]. Chief among the benefits of long-acting injectable drugs is that they circumvent the commonplace problem of ensuring, over months to years of time, compliance with the frequent dosing required by oral regimens. In previous studies with *P. berghei* ANKA in mice, we found that an oral dose of atovaquone protected for one day, but when the same dose was given as a single injection of atovaquone nanoparticles in aqueous suspension, the protection lasted four weeks, a duration expected to be longer in humans[8].

Reflecting the complex lifecycle of malaria parasites within humans (each stage with its distinctive biochemistry and pharmacology), conventional antimalarial drugs have limited scope in human infections. Although it is the sporozoite form that is transmitted by female anopheline mosquitoes, the parasites of critical importance for chemoprophylaxis against symptomatic disease have been the initial (and entirely asymptomatic) liver stage, and the subsequent asexual (pathogenic) erythrocytic stages – targets for causal and suppressive prophylaxis, respectively. Most agents are suppressive (e.g., chloroquine, mefloquine, doxycycline) but the synergistic fixed combination of atovaquone plus proguanil has both causal and suppressive action. Key among the advantages of causal prophylaxis is the substantially smaller number of parasites (difference ≥4 logs) that must be overcome, which reduces the likelihood of failure or resistance.

Atovaquone was devised as a structural mimic of coenzyme Q (or ubiquinone, an essential component in mitochondrial electron transfer) that binds to cytochrome *b*, collapses the transmitochondrial membrane potential, and impairs the malaria parasite's obligatory synthesis of pyrimidines[9,10]. Atovaquone has garnered a reputation for being safe and well-tolerated when used alone for prophylaxis or treatment of *Pneumocystis jirovecii* or *Toxoplasma gondii* infection, which entails months to years of high dose therapy and plasma levels logs greater than those required for causal malaria prophylaxis[9]. Development of atovaquone alone to treat established erythrocytic malaria infections was abruptly paused by the unexpectedly ready appearance of drug resistance, mediated by mutations in *cytB*, the mitochondrial gene encoding cytochrome *b*, a finding that confirmed the mechanism of action[11,12]. In an effort to salvage atovaquone, selected from among several possible partners was proguanil, itself also active against liver-stage parasites and substantially synergistic with atovaquone. Clear reductions in requisite dose of each partner drug against erythrocytic parasites was shown in laboratory and clinical studies[11–13]. Field studies with atovaquone + proguanil confirmed reliable cures[14], and the causal prophylactic activity of atovaquone alone was demonstrated in humans[15]. Accordingly, fixed combination atovaquone + proguanil was FDA approved in 2000 for the treatment of malaria. Although the impact of co-administered proguanil on efficacy and resistance against liver stages was not assessed, the same formulation was also approved for causal prophylaxis, making this a notable exception to the usual single drug malaria prophylaxis.

Numerous studies have focused on the issue of atovaquone resistance, pinpointing mutations in loops of cytochrome *b* that extend into the intermembrane space, where reduced coenzyme Q binds. Mutants have been obtained in vivo (from clinical drug treatment failures or murine models) and from in vitro selection. Curiously, there are marked differences in the mutants obtained clinically *versus* in vitro. The several dozen literature reports of atovaquone + proguanil treatment failures for *falciparum* malaria all document highly resistant Y268N/S/C mutants[16]. (Reported without drug sensitivities are I258M[17] and no mutation in *cytB*[18,19]). By contrast, in vitro experiments with *P. falciparum* have generated at least 11 different mutations, some multiple, all in the intermembrane space or transmembrane regions of the protein, and with just one exception[20], none at Y268[12,20,21]. Furthermore, atovaquone resistance of in vitro-generated mutants is

substantially lower (typically 100- to 1000-fold) than that of the clinically relevant Y268N/S/C. The dichotomy between in vitro and clinical mutants assumed greater importance with intriguing reports that the development of atovaquone-resistant parasites in mosquitoes is impaired or halted[22–24], suggesting their transmission may be limited in the field. Unfortunately, these studies have not included an evaluation of the Y268 mutants of *P. falciparum* that pertain to clinical drug resistance.

Transmissibility, or not, of atovaquone resistance is clearly critical to the use of long-acting injectable atovaquone as a chemical vaccine against malaria. In this study we generated a clinically relevant Y268S mutant of *P. falciparum* NF54, and, in comparison to isogenic wild type (WT) control parasites, monitored its growth and differentiation in anopheline mosquitoes and its transmissibility to mice engrafted with human hepatocytes and erythrocytes.

## Results

### Genesis and in vitro characterization of atovaquone-resistant *P. falciparum*

We were unable to obtain from other sources atovaquone-resistant *P. falciparum* with a Y268 mutation in cytochrome *b*, that retained the ability to infect mosquitoes. Accordingly, they were generated in vitro by applying atovaquone pressure to mosquito-competent *P. falciparum* NF54 lines, minimally passaged after isolation from clinical trial volunteers.

Over a 21-month period, efforts by several investigators, entailing a total of $7.8 \times 10^9$ gametocyte-competent parasites, yielded five independent isolates of atovaquone-resistant cells, three of which were Y268S mutants. To select these mutants, continuous drug pressure was applied to various parasite densities and hematocrit levels. Mutants were detected by microscopy 25–47 days after start of pressure (Supplementary Fig. S1a, b). Reasoning that the mutation rate in mitochondrially-encoded *cytB* may be greater in cells cultured in a relatively high oxygen atmosphere[25], selections (and parallel no drug controls) were conducted with cells acclimated to conventional incubator conditions of 5% $CO_2$ in air (i.e., 19% $O_2$)[26]. This is substantially higher than the 2–3% $O_2$ in classic cultivation of *P. falciparum*[27], or the ~13% $O_2$ in our routine candle jar conditions, but it nevertheless falls within the range experienced by parasites in vivo[26]. Below-described studies in mosquitoes and mice revealed no difference between parasites maintained continuously in candle jar conditions *versus* those in 5% $CO_2$ in the air for the ~two-month period of resistance selection.

Parasites that emerged from drug pressure were first evaluated for changes in *cytB*, by PCR amplification and sequencing of the entire open reading frame (Supplementary Fig. S2a, b). Three independent mutants encoded a Y268S amino acid change (Supplementary Fig. S1b, c). Whole genome sequence analysis of mutant cells was obtained at a coverage of 44- to 57-fold and compared to the reference *P. falciparum* NF54 gene sequence in PlasmoDB. FreeBayes variant caller in Partek Flow identified the Y268S mutation with a Variant Quality (VarQual) score of 49,191. All other genetic differences were of low relevance with VarQual scores of 721 or less (Supplementary Data File 1).

These collective efforts provided a mutation rate of $6 \times 10^{-10}$. This is somewhat less than the previously reported $4.6 \times 10^{-9}$ for atovaquone against 3D7[28], and substantially lower than $10^{-5}$ reported for atovaquone against multi-drug resistant W2 from Indochina[29], perhaps reflecting our use of high drug concentrations for selection, or perhaps there is some impact of the retained (and likely costly to the cell) ability of these parasites to generate infectious gametocytes. Interestingly, all three Y268 mutations were Y-to-S (TAT-to-TCT), as was the only other previously reported in vitro-selected Y268 mutant[20]. We did not obtain the Y268N (to-AAT) or Y268C (to-TGT) mutants that are also seen clinically. Although all four known in vitro-selected mutants, from two laboratories, are thus Y268S, and this may have happened by chance,

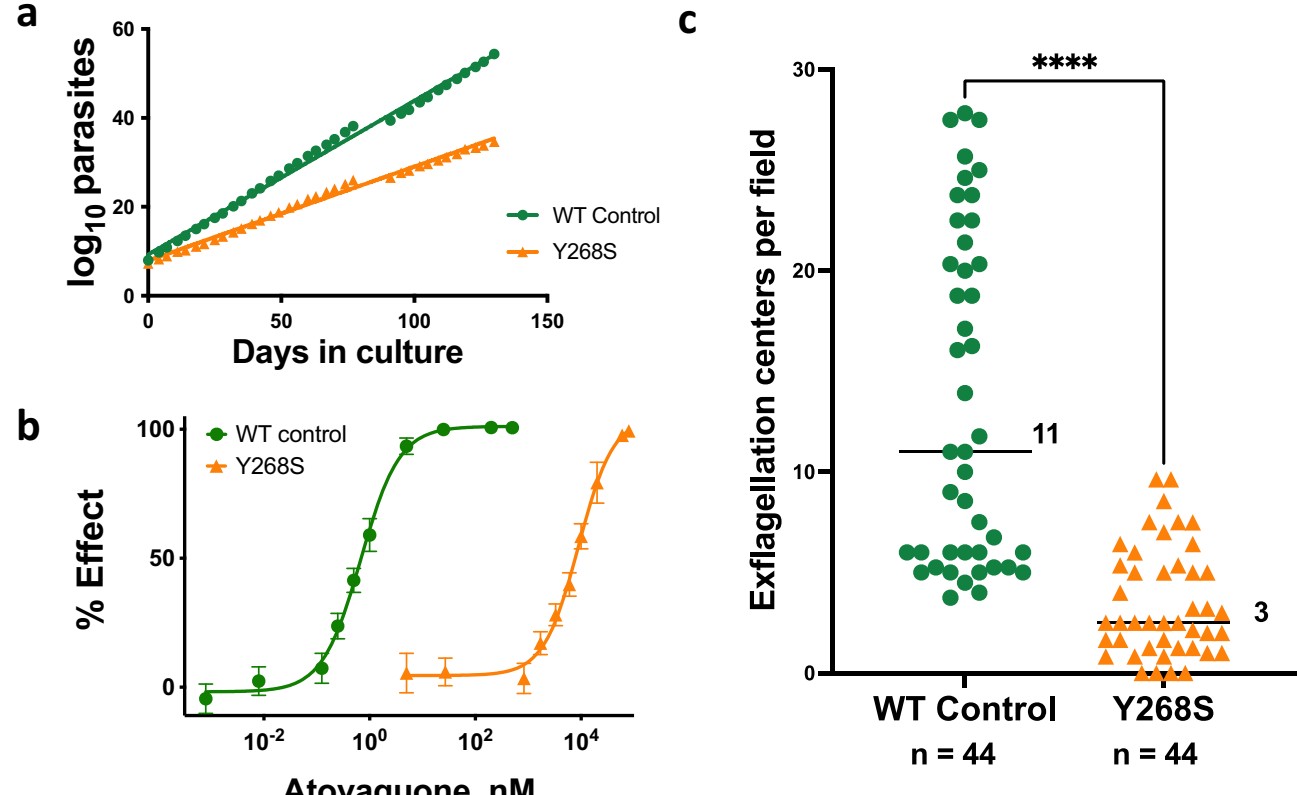

**Fig. 1 | Phenotype characterization in vitro of isogenic wild type (WT) and cytochrome *b* Y268S mutant *P. falciparum*. a** Cumulative number of asexual parasites in continuous culture. Growth rate of wild type parasites (y = 0.35x + 9.3) was 1.4-fold greater than that of mutant (y = 0.21x + 7.9); *n* timepoints = 36, $R^2 > 0.99$, data from one continuous experiment. **b** Atovaquone activity against WT or Y268S asexual erythrocytic parasites ($EC_{50}$ 0.68 nM or 9.0 μM, respectively) was obtained by assaying [³H]hypoxanthine incorporation as a function of atovaquone concentration. Depicted are mean ± SD of quadruplicate determinations from each

of three biological replicate experiments (for each data point *n* = 12; some SD are too small to extend outside the symbols); $R^2 \geq 0.994$. **c** Male gametocyte exflagellation, adjusted to 1.5% gametocytemia. *Bars*, median number of centers across four independent biological replicates (ranges 4 to 28 for WT, and 0 to 10 for mutant); *n*, the total number of fields examined; ****$P < 0.0001$, two-sided Mann–Whitney test, single comparison. For WT cells, all 44 microscopy fields had at least one exflagellation center (100%); Y268S evidenced exflagellation in 40 of 44 fields (91%). Source data are provided as a Source Data file.

failure to obtain Y268N/C may also reflect a biological bias that operates for selection in vivo but not in vitro.

Asexual erythrocytic Y268S parasites had several phenotypic changes. Their growth rate in vitro was only 71% that of isogenic parent cells (Fig. 1a), and they reached lower peak parasitemia (Supplementary Fig. S3a), findings consistent with a fitness cost for this mutation. Compared to WT, mutant parasites had significantly fewer progeny within late schizonts, a difference that almost certainly contributes to slower growth (Supplementary Fig. S3b). However, the *cytB* mutation and atovaquone resistance remained stable for at least four months in the absence of drug pressure, indicating the fitness cost is sustainable for asexual erythrocytic parasites in vitro.

As expected, the Y268S mutation was associated with substantial atovaquone resistance (Fig. 1b). Baseline $EC_{50}$ for wild type control parasites was 0.68 nM, in line with published values of less than 1 nM[30], but was 9.0 μM for the Y268S mutant, an impressive change in drug susceptibility also in keeping with previous reports[12, 31]. Preliminary experiments in our hands indicated that $EC_{50}$ values for atovaquone can vary two- to ten-fold depending on the protein source in the medium (albumax *versus* human serum) or differential count of the starting culture. These, in conjunction with atovaquone's limited aqueous solubility at the highest assay concentrations, likely contribute to the 75-fold range of $EC_{50}$ values reported by different labs for Y268S clinical isolates[12,31,32]. An ambient atmosphere (3% $O_2$, candle jar, or 5% $CO_2$ in air) had no significant effect on $EC_{50}$ values. Susceptibilities to artemisinin (9.8 *versus* 12 nM) or chloroquine (8.2 *versus*

11 nM) were not significantly affected by the cytochrome *b* Y268S mutation.

The final in vitro assay we conducted was an assessment of male gametocyte exflagellation (Fig. 1c). This phenomenon normally occurs in the midgut of anopheline vectors after ingestion of an infected bloodmeal, during which male gametocytes release up to eight motile gametes[33]. This process, induced in vitro by incubating mature gametocytes at room temperature[34], revealed a highly significant nearly four-fold decrease in exflagellation centers in Y268S mutants as compared to isogenic controls, likely reflecting an increased demand for mitochondrial respiration[35].

### Characterization of parasites in anopheline mosquitoes

In all experiments, stage V gametocytemia in infecting bloodmeals was adjusted to be the same, at 0.5%. Ookinetes (both immature and mature) were evaluated in the midguts of blood-fed *Anopheles stephensi* at 20–26 h after infection. The count in mosquitoes infected with WT parasites was nearly eight-fold greater than that in Y268S mutant-fed insects (Supplementary Fig. S4a, c, d).

Oocyst numbers on *An. stephensi* midguts, evaluated in nine independent experiments (Fig. 2a), revealed a highly significant difference in the number of wild type or isogenic Y268S mutants (medians 39 or 0 per midgut, respectively). Prevalence of mutants was similarly reduced (Fig. 2a, pie charts). Furthermore, although drug-resistant Y268S parasites generated up to five oocysts/midgut in 15% of samples, mercurochrome-stained images revealed that their quality

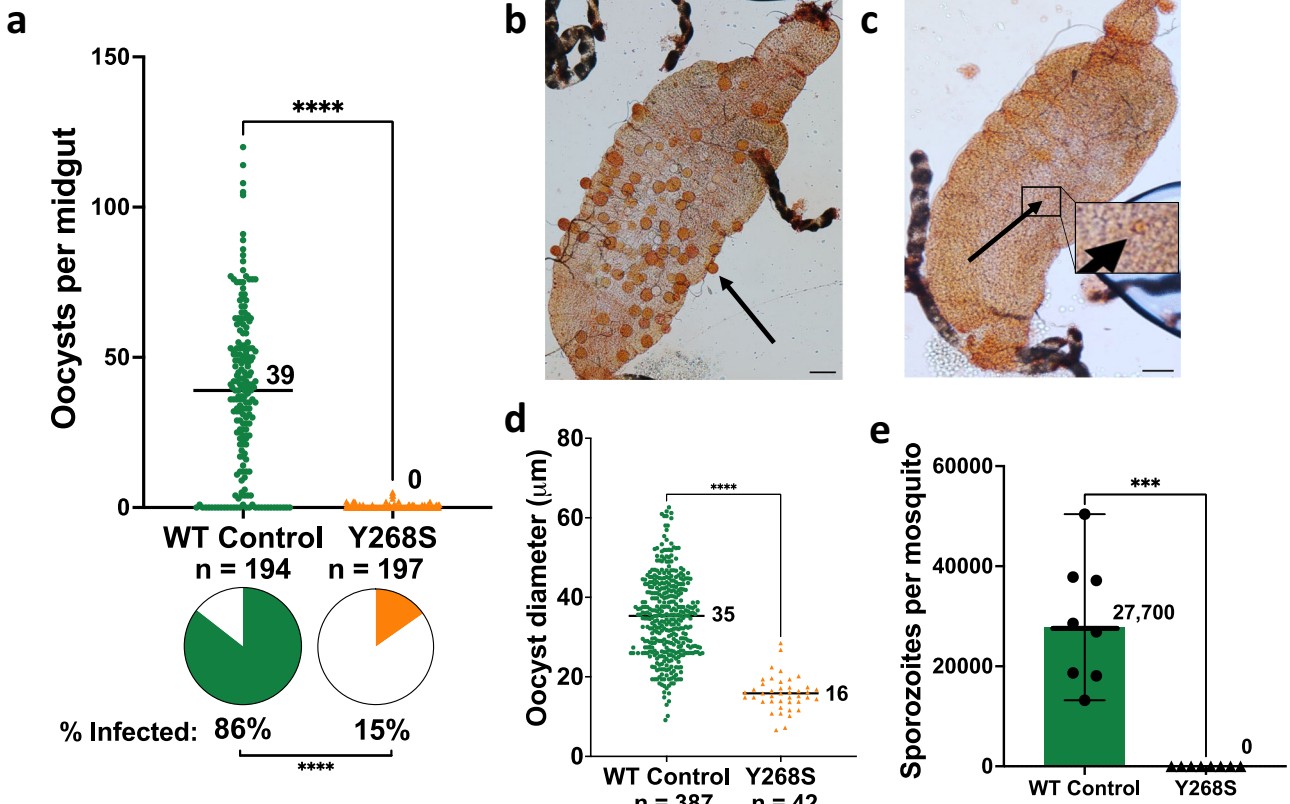

**Fig. 2 | Evaluation of wild type and Y268S *P. falciparum* in *An. stephensi* tissues.** All feeds were adjusted to 0.5% stage V gametocytemia. **a** Oocysts on mosquito midguts were counted at 9-10 d after membrane feed. Indicated is the median number of oocysts per midgut in nine biological replicate experiments (ranges 0 to 120 for WT, and 0 to 5 for Y268S); *n*, the total number of mosquitoes dissected, two-sided Mann–Whitney test, single comparison; *pie charts*, percent of mosquitoes infected; ****P < 0.0001, two-sided Fisher's exact test. **b**, **c** Photomicrographs of representative midguts infected with WT or Y268S parasites, respectively. *Bars*, 100 μm; *arrows*, representative oocysts; *inset*, magnification of Y268S oocyst; banded black structures are Malpighian tubules. **d** Oocyst diameters were measured in six biological replicates for wild type, and seven biological replicates for Y268S parasites. Median diameter is indicated (WT range 9 to 63 μm, mutant range 7 to 28 μm); *n*, the total number of oocysts measured; ****P < 0.0001, two-sided Mann–Whitney test, single comparison. **e** Salivary glands were harvested at 17–20 d after feed in eight biological replicate experiments entailing a total of 234 WT-fed or 245 Y268S-fed mosquitoes. *Symbols*, for each experiment, average number of sporozoites per mosquito. No sporozoites were seen in any Y268S preparation. *Bars*, median (values indicated) with 95% confidence interval; ***P = 0.0002, two-sided Mann–Whitney test, single comparison. Source data are provided as a Source Data file.

was uniformly poor, appearing smaller and malformed relative to their WT parents (Fig. 2b, c), with median diameter less than half that of WT (Fig. 2d).

*An. stephensi* salivary glands were examined for sporozoites in eight biological replicate experiments (Fig. 2e). The difference between WT and Y268S mutant parasite counts was stark: in a total of nearly 250 mosquitoes per group, median counts were 27,700 and zero, respectively. The difference in the maturation time of mutant *versus* wild-type sporozoites as a possible cause of these discrepant counts was ruled out (Supplementary Fig. S5).

To test whether the phenotypic differences between wild-type and atovaquone-resistant parasites in *An. stephensi* would apply in another major malaria vector, we repeated these experiments with *An. gambiae*, and obtained essentially the same results: ookinetes were less (Supplementary Fig. S4b, e, f), and median counts were reduced from 22 to 0 for oocysts and from 2700 to 0 for sporozoites/mosquito (Supplementary Fig. S6). As with *An. stephensi*, no sporozoites were seen in any of the Y268S salivary gland preparations.

Since malaria infections in the field are frequently polyclonal, we wanted to determine whether the severe impairment in Y268S sporozoite number could be rescued by coinfection with wild-type parasites. *An. stephensi* were fed gametocyte cultures with WT *P. falciparum* only or with mixtures of 1:1 or 1:10 WT:drug-resistant parasites. In three biological replicates, median oocyst numbers were 11, 5, and 1 for WT,

1:1, and 1:10 groups, respectively (Supplementary Fig. S7a). Sporozoite counts were similarly patterned (Supplementary Fig. S7b). Thus, although total gametocyte numbers in the feeds were constant, the number of tissue parasites, for both oocysts and sporozoites, correlated strongly with proportion of WT gametocytes in the bloodmeal (Supplementary Fig. S7c). In all three replicate experiments, these finding were further confirmed by two complementary RFLP assays, designed to detect 10% or less mutant *cytB* DNA in midgut or salivary gland samples from mixed infections (Supplementary Fig. S2d). All samples evidenced WT DNA, but no mutant DNA was detected in any of the mixed infection samples (Supplementary Fig. S7d). These findings suggest not only that the cost of the Y268S mutation in mosquitoes is not relieved by coinfection with WT gametes, but indeed that WT parasites may outcompete the mutants in a mixed population.

## Transmissibility of *P. falciparum* from mosquito to humanized mouse

To provide a more biologically relevant, and functional, assay for the transmissibility of the mutant parasite, huHep mice were bitten by *An. stephensi* infected with either WT or Y268S atovaquone-resistant *P. falciparum* (Supplementary Fig. S8). In two independent experiments, three mice were bitten by an average of 138 wild-type-infected mosquitoes/mouse and two were bitten by an average of 152 mutant-infected mosquitoes/mouse (median parasite intensities in these

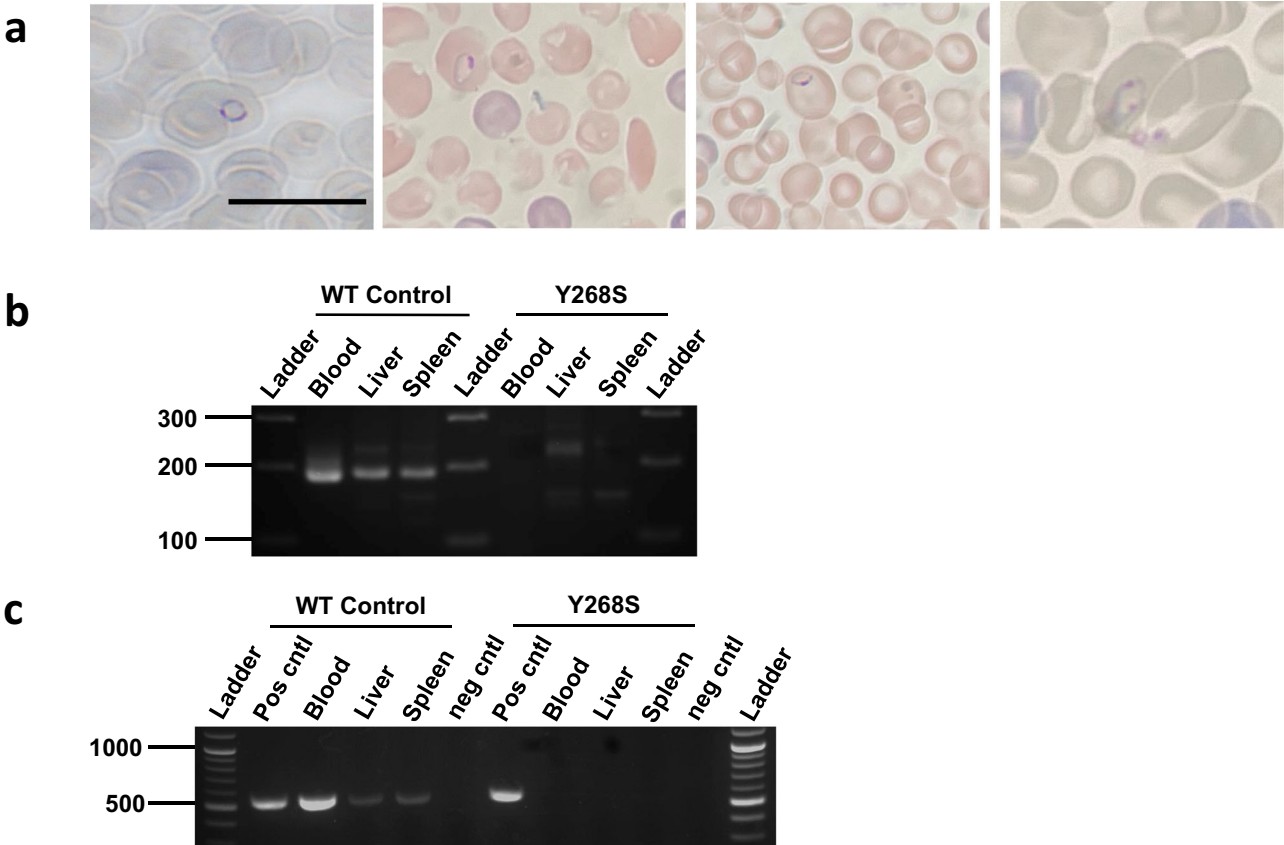

**Fig. 3 | Analysis of huHep mouse tissues for evidence of WT or Y268S *P. falciparum*. a** Giemsa-stained thin smears of peripheral blood from mice bitten 7 to 7.5 d previously by *An. stephensi* infected with WT parasites. Intracellular ring stage parasites are evident within the engrafted human erythrocytes, which at 6.4 μm are appreciably larger than the endogenous 4.7 μm mouse cells. Samples are from two mice in each of two biological replicate experiments; *bar*, 10 μm. Analogous peripheral blood from mice bitten by Y268S-infected mosquitoes evidenced no parasites in ≥16,000 RBCs examined per mouse, by at least two independent observers. **b** Indicated tissues were harvested from mice bitten 7.5 d previously by mosquitoes fed on WT or Y268S 0.5% gametocyte culture; liver and spleen were obtained after exsanguination and saline perfusion. DNA was isolated and analyzed by nested PCR for nuclear DNA-encoded *STEVOR* sequence (188 bp product). *Ladder*, indicated size fragments. Tissues were obtained in two biological replicate experiments and PCR assays were done in two or more independent technical replicates. **c** In a second independent experiment, tissues were harvested as above, and isolated DNA was analyzed by nested PCR for mitochondrial DNA-encoded *cytB* sequence (538 bp product). *Ladder*, indicated size markers; *pos cntl*, template DNA isolated from cognate asexual parasite culture; *neg cntl*, no template added reaction. (Longer exposures of Panels B and C are in Supplementary Fig. S10.) Tissues were obtained in two biological replicate experiments and PCR assays were done in two or more independent technical replicates.

mosquitoes were comparable to overall data in Fig. 2a, e: 29-33 WT or zero Y268S oocysts/midgut; 18,000 WT or zero Y268S sporozoites/mosquito). Engraftment of human erythrocytes assayed after infusions on days 5.5 and 6 after infection, was robust (WT and mutant averaged 30 and 29%, respectively). By 7.5 d after infection, all mice bitten by WT-infected mosquitoes were parasitemic (0.18-0.26%) on Giemsa-stained blood smear (Fig. 3a), as confirmed by blood culture. However, no parasites were seen in smears from mice bitten by mutant-infected mosquitoes, and in vitro cultures of ≥1 mL blood samples harvested at necropsy remained parasite-free for at least three weeks. Confirming these results, blood, liver and spleen from mice infected with WT parasites were positive for both *STEVOR* and *cytB*, but samples from mice exposed to Y268S mutant *P. falciparum* were negative by PCRs (Fig. 3b, c).

## Discussion

Atovaquone is, in many ways, a uniquely ideal candidate for a long-acting injectable chemical vaccine for malaria. It is potent against both initial liver and pathogenic erythrocytic parasites, for both *P. falciparum* and *P. vivax*; it has a decades-long track record of safety in patients whose plasma drug levels are 100 times higher than those required for causal activity; it is not metabolized and has few drug-drug interactions; and it faces no established resistance in the field or recognized cross-resistance with other antimalarials. No other currently available antimalarial even approaches these characteristics. Exciting candidates in development[2] still require years of nonclinical study and early clinical trials, none of which can provide the extensive human exposure that bears on safety. More practically, further development as a chemical vaccine is facilitated by the fact that atovaquone (alone as Mepron, or in fixed combination with proguanil as Malarone) is off-patent, which, together with extensive use in humans, streamlines its regulatory requirements and facilitates its more equitable deployment.

Countering these advantages is a concern about atovaquone resistance. Selected readily when symptomatic patients harboring $10^{9-12}$ erythrocytic parasites are treated with atovaquone alone, or occasionally in such patients treated with atovaquone + proguanil, clinical drug-resistance is mediated almost exclusively by a single point mutation at the Y268 residue in cytochrome *b*. In the setting of long-acting atovaquone monoprophylaxis, resistance concerns are twofold. *First* is the possibility that causal therapy may itself select resistant parasites. Several factors mitigate against this, including the fact that drug is present before any parasites are introduced, and that the number of liver stage parasites from which to select a mutant is

relatively small (by orders of magnitude, a recognized "bottle neck" in the lifecycle). This analysis is supported by the absence of reported resistance in people prophylaxed with atovaquone + proguanil[36], despite its being the most widely prescribed agent for this indication[37], and the tens of millions who travel to malarious areas every year[38]. *Second* is the concern that mutants occurring in a person dosed with long-acting atovaquone will be transmitted by mosquito to others. Importantly, atovaquone + proguanil is not standard therapy for malaria patients in the field, and there is no cross-resistance with other drugs. In the unlikely event of mosquito-borne resistance, the consequence would not be the loss of available therapies to treat the affected patients but rather the possible loss of atovaquone as a chemical vaccine.

Our studies focused on the transmissibility of the clinically relevant highly atovaquone-resistant cytochrome *b* Y268S mutant *P. falciparum*. In 14 independent experiments, including both *An. stephensi* and *An. gambiae* mosquitoes, by conventional light microscopy and with multiple experienced observers, we found not one Y268S sporozoite in salivary glands from 345 mosquitoes. Validating this finding, the functional bioassays with huHep mice were especially important. By multiple criteria, including light microscopy (tail snip blood films), exquisitely sensitive PCR for *P. falciparum* nuclear or mitochondrial genes (liver, blood, spleen), or in vitro culture (of total blood volume), they evidenced no transmission of Y268S parasites. Furthermore, a mixture of mutant and wild-type parasites, to mimic polyclonal infections available to mosquitoes in the field, failed to boost sporozoite numbers above those expected for the WT component alone, and instead suggested that wild type parasites out-compete the mutant. In our hands, the fitness cost of the clinically relevant Y268S mutation in cytochrome *b* is severe-to-lethal in the mosquito.

At least three other laboratories have evaluated the survival in mosquitoes and in some cases transmissibility, of atovaquone-resistant parasites[22–24]. None of these reports include a clinically relevant *P. falciparum* Y268 mutant. Previous studies included *P. falciparum* with 5 to 20-fold resistant M133I or V259L, and *P. berghei* with both low and high resistance (M133I, Y268C, Y268N). We also isolated a low-grade L144S mutant, and found that it, like previously reported M133I, generated sporozoites (Supplementary Figs. S1b, S5, S9). The best interpretation for all these studies appears to be that laboratory-generated malaria parasites (including *P. falciparum*) with low-level atovaquone resistance can produce microscopically-detectable sporozoites in mosquitoes, whereas parasites with clinically relevant high-grade resistance cannot. Of obvious interest is the relevance of these various mutants to the use of atovaquone in humans.

The striking difference in EC$_{50}$ values seen for atovaquone-resistant parasites obtained in vitro *versus* those arising in treated humans is almost certainly related to the disparate drug concentrations applied for selection. Atovaquone levels applied in vitro have not exceeded 100 nM, whereas trough plasma concentrations are 5400 nM (2 µg/mL) in subjects dosed daily for malaria prophylaxis[39], and are even higher in those treated for infection. This in turn suggests that parasites with low-level resistance will not survive in treated humans. In conjunction with our observation that the clinically relevant, highly resistant Y268S fares poorly, if at all, in mosquitoes, these findings mitigate concern about transmissibility of atovaquone-resistance.

The availability of gametocyte-competent wild-type *P. falciparum* and isogenic mutants with various degrees of resistance provides a new resource for further studies. For example, with highly resistant Y268S we see fitness costs throughout the lifecycle, starting with a 29% reduced growth rate of asexual erythrocytic stages, and then a progressive loss in the mosquito: 73% reduction in exflagellation, 83% reduction in oocysts, and complete loss of sporozoites (Figs. 1 and 2). By contrast, sporozoite numbers in our L144S mutant are reduced 63%, but not eliminated (Supplementary Fig. S9e). These findings, in

parasites that differ by just a single amino acid residue in cytochrome *b*, suggest that fitness in the mosquito may be related to the degree of atovaquone resistance. On a molecular basis, a relationship between the degree of resistance and fitness implies that mutations in cytochrome *b* that impair atovaquone binding correspondingly impair catalysis. Highly resistant asexual erythrocytic parasites evidence a mild but detectable growth defect, in keeping with their limited dependence on mitochondrial respiration. However, in the mosquito where mitochondrial function is essential[22], even low-level resistance markedly impacts growth, and high-level resistance is lethal. These findings are akin to previous reports of differences in the pattern of *P. falciparum* drug-resistance polymorphisms in human blood *versus* mosquito tissues, likely also reflecting dissimilar selective pressures in the hosts, that may impose vector-selective constraints on drug resistance in the field[40].

Our studies indicate that atovaquone resistance may not pose as great a threat to long-acting causal prophylaxis as it did historically to the treatment of established erythrocytic infection, and that clinically relevant highly resistant *P. falciparum* mutants are not transmitted by mosquitoes. In view of resurging malaria numbers, limited efficacy and durability of current biological vaccines, and the need for new control strategies, it would seem sensible to establish proof-of-concept for atovaquone as a chemical vaccine. No amount of work can prove the negative (*i.e.*, resistance is not transmissible) and no laboratory investigations can address the enormous diversity of real-world malaria. Despite the promise of our studies, transmissibility, or not, of atovaquone resistance will best be determined by carefully designed and conducted clinical trials in endemic areas.

## Methods

### *P. falciparum* cultivation and mutant selection

Studies were conducted with low passage *P. falciparum* NF54 (MRA-1000 BEI Resources, NIAID, NIH), grown asynchronously at 37 °C under candle jar atmosphere ( ~ 13% O$_2$, 3% CO$_2$, 84% N$_2$) in RPMI supplemented with 2.1 mM L-glutamine, 25 mM HEPES, 0.25% NaHCO$_3$, and 0.37 mM hypoxanthine, in 10% heat-inactivated human serum (Interstate Blood Bank) and O+ erythrocytes at 2 or 4% hematocrit[27]. Periodic testing for mycoplasma was negative (MP0025, Sigma-Aldrich). Erythrocytes obtained weekly from healthy donors under a Johns Hopkins Institutional Review Board-approved protocol were provided without identifiers to the laboratories. Prior to atovaquone pressure, parasites were adapted over 14 d to 5% CO$_2$ in air, as described previously[26].

For mutant selection, atovaquone (PHR1591, Sigma Aldrich) in DMSO (D128500, ThermoFisher) was further diluted in a medium containing 10% serum such that the final DMSO was ≤0.1%. Cultures were exposed in parallel to atovaquone or no drug (Supplementary Fig. S1a,b). Medium was replaced every other day; cultures were split 1:2 every 7 d; only the first split was retained. Cells were maintained under constant drug pressure for 60 days or until healthy parasites appeared. Upon confirmation of Y268S mutation, atovaquone was removed and drug-resistant lines and their concurrent no-drug controls were returned at once to candle jar conditions, amplified, and aliquots were frozen (Glycerolyte 57, McKesson General Medical). In all subsequent experiments WT and mutants were manipulated and studied identically and in parallel.

To obtain gametocytes, cultures were initiated at 0.5% parasitemia, and 4% hematocrit, and the medium was replaced daily for 15-19 d until gametocytes reached stage V. Mutant parasites are being deposited with ATCC.

### In vitro drug sensitivities

Asynchronous *P. falciparum* (0.25% parasitemia, 1.2% hematocrit) in microtiter plates was exposed to serial dilutions of the drug (in quadruplicate) for 72 h. [³H(G)]-hypoxanthine (16 Ci/mmol, 1 mCi/mL NET177001MC, Perkin Elmer) was added for the final 24 h, and

incorporated radiolabel was evaluated in harvested cells, as described previously[41]. $EC_{50}$ values were obtained by nonlinear regression analysis (Prism 6; GraphPad).

## cytB and whole genome sequences

For *cytB* sequence in parasites surviving drug pressure, lysates were prepared from cultures at 1% parasitemia (90 °C, 5 min), and an 1848 bp fragment (containing the 1131 bp *cytB* open reading frame) was amplified with primers (Supplementary Fig. S2a, b) and Phusion DNA polymerase (NEB). PCR product was verified by gel electrophoresis as single band of predicted size, and sequenced. Sequences were aligned to *P. falciparum* 3D7 cytochrome *b* (PF3D7_MIT02300) using NCBI blast, and positions on all trace files were assessed in SnapGene 7.0.

For whole genome sequences to identify the causative mutation[42], DNA was isolated from $10^8$ asexual parasites; QiAamp DNA Mini Blood kit, Qiagen), quantified (Qubit HS ds DNA assay (Qubit Flex Fluorometer, ThermoFisher)), and quality assessed by Genomic Screentape analysis (TapeStation 2200, Agilent). Barcoded libraries for DNA-Seq were synthesized from 100 ng DNA (Celero EZ-Seq kit with NuQuant, Tecan Genomics), quality assessed by High Sensitivity DNA Lab Chips (BioAnalyzer 2100, Agilent), quantified by Qubit HS dsDNA assay, and sequenced on Illumina's MiSeq platform, (2 x 300 bp v3 with 5% PhiX; JHMI Synthesis and Sequencing Facility). Using Partek Flow 10.0.22.0410 with sequence defaults, analyses included pre-alignment QA/QC, adaptor/read trimming, reference genome alignment to *P. falciparum* NF54 (PlasmoDB) using Bowtie2 2.2.5, post-alignment QA/QC, FreeBayes variant calling, variant filtering (VarQual ≥ 30), and variant annotation based on precedence rules. To identify changes unique to atovaquone-resistant parasites, sequences were compared to that of the reference strain *P. falciparum* NF54 GCA009761475.1 (PlasmoDB).

## Gametocyte exflagellation

Just prior to mosquito feed, 200 μL of gametocyte culture was taken to assess male exflagellation[43,44]. Briefly, cultures were centrifuged in a warm tube (500 x *g*, 5 min), 2 μL of cell pellet was placed on a glass slide, covered, and incubated (room temperature, 15 min). The number of exflagellation centers in 11 independent 400X fields was counted and normalized to 1.5% gametocytemia. In three of four experiments, counts were obtained in a blinded fashion.

**Mosquito maintenance and infection.** All experiments were conducted with *An. stephensi* (Liston, NIH) or *An. gambiae* (Keele, NIH) reared and maintained in the Johns Hopkins Malaria Research Institute insectary (27 °C, 80% humidity, 12/12-h light/dark cycle), and provided 10% sucrose solution-soaked cotton pads[43]. For each experimental group, 100–200 mosquitoes at 3–7 d post-emergence were fasted overnight prior to 37 °C water-jacketed glass membrane feeding, for one hour, by established protocols[43]. All bloodmeals were adjusted to 0.5% stage V gametocytemia. Thereafter, 10% sucrose was withheld for 48 h to eliminate unfed mosquitoes.

## Ookinete, oocyst and sporozoite counts

Ookinetes were counted by an adaptation of established protocols[45]. At 20–26 h post bloodmeal, midguts were dissected from ten visibly blood-fed mosquitoes, placed in 20 μL 3% acetic acid, and vigorously pipetted to lyse midguts. An aliquot (2 μL) was loaded on a hydrophobic slide (VWR, 100488–892), spread to fill a 14 mm circle, dried, fixed with methanol, and stained with Giemsa. For each experimental group, the number of mature and immature ookinetes in sixty 1000X fields was recorded.

Oocyst and sporozoite counts were obtained by adaptation of established protocols[43]. At 9–10 d post feed, 15–30 midguts per group were dissected into 1X PBS on a hydrophobic slide (VWR, 100488-892). Samples were stained with 0.1% mercurochrome (M7011, Sigma Aldrich), covered, and examined (100X) for oocysts. Oocyst diameters

were analyzed in ImageJ version 1.53. At 17–20 d post feed, salivary glands from 20-38 mosquitoes per group were collected, transferred to 100 μL 1X PBS on ice, sedimented (5000 *g*, 30 sec), and homogenized (sterile pestle, 30 sec). Parasites were counted by hemocytometer to obtain an average sporozoite count per mosquito.

## Transmission from *An. stephensi* to huHep mice

All animal studies were conducted under a protocol approved by the Johns Hopkins Animal Care and Use Committee with huHep mice (FRG KO (Fah–/–, Rag2–/–, Il2rg–/–) on NOD background) engrafted with human hepatocytes and capable of maintaining circulating human erythrocytes (Yecuris[46,47]). Female mice (27–28 wk old) were housed with sterile water, chow (*ad libitum*), and bedding, at 20-23 °C and 30-70% humidity, with 12-hr light/dark cycles. Until ≥8 d prior to infection they were maintained on 1 mg/mL 2-(2-nitro-4-trifluoromethylbenzoyl)1,3-cyclohexedione (CuRx™ Nitisinone, 20-0027) in sterile 10% dextrose. Given the rarity and expense of huHep mice, the subtle, if at all, difference in infectivity of female *versus* male animals (including humans), these studies were conducted with female mice. Mosquitoes (17–18 d post-infection with WT or isogenic atovaquone-resistant parasites) were fed through a mesh-covered container on mice anesthetized with 4–6 μL/g ip of 14 mg/mL ketamine (67457010810, Myland Pharmaceuticals) plus 1.2 mg/mL xylazine hydrochloride (X1251-1G, Sigma Aldrich) in PBS. Mosquitoes fed for 14 min, with interruption every two min to limit blood loss in mice; over 80% evidenced having taken blood. At 5.5 and 6 d post mosquito probe, mice were injected intravenously with 600 μL human blood (50% red cells, 25% O+ serum, 25% RPMI). Human erythrocyte engraftment was assayed by flow cytometry of peripheral blood incubated 1:50 with APC-conjugated TER-119 rat antimouse monoclonal antibody (BD Biosciences catalog no 557909, lot 9213638), with gates at $APC^+$ and $APC^-$ (mouse and human erythrocytes, respectively) (Attune NxT Flow Cytometer, BD Biosciences) (Supplementary Fig. S11).

Starting 6.5 d after infection, Giemsa-stained blood films were monitored twice daily. At patency, all mice were sacrificed. Blood was collected by cardiac puncture into citrate/phosphate/dextrose (Sigma, C7165), for PCR (150 μL aliquot) and in vitro culture (the ≥1 mL residual blood volume). After perfusion with 15 mL 0.9% NaCl, the liver and spleen were harvested, weighed, and stored at −70 °C. Blood cultures were maintained for three weeks or until a robust parasitemia was evident.

## PCR assays

Mouse tissues were assayed for *STEVOR* or *cytB* by nested PCR. DNA was isolated from blood (QiAamp DNA Mini Blood Kit, Qiagen; or Monarch Genomic DNA purification kit, NEB) or from perfused organs homogenized (Fisher Bead Mill 4, 2.4 mm metal beads) prior to extraction (DNA Maxi Kit; Qiagen). For *STEVOR* assay, eluates containing DNA (from 25 μL whole blood, or 2 μg DNA from liver or spleen (Nanodrop 2000, ThermoFisher)) were analyzed as described previously[48,49]. For *cytB* analysis, the 25 μL primary amplification reaction contained 50 nM each primer (Supplementary Fig. S2a,c), 200 μM each dNTP (NEB), 1 U Phusion DNA polymerase (NEB), and 10 ng template DNA. The 25 μL nested reaction contained 200 nM of each primer (Supplementary Fig. S2a,c), 200 μM each dNTP, 1 U of Phusion DNA polymerase, and 1 μL of the primary reaction product. Thermocycling conditions for both reactions were: denaturation (98 °C, 3.5 min); 25 cycles of (denaturation (98 °C, 30 sec), annealing (56 °C, 40 sec), and extension (72 °C, 45 sec)); and final extension (72 °C, 5 min). Products were separated by agarose gel electrophoresis and visualized by ethidium fluorescence. Serial dilutions into whole blood of a known number of *P. falciparum*-infected erythrocytes indicated both methods detect ≥10 parasites per 25 μL blood sample.

## Statistical analyses

Data were collected in Excel 16.73. Dose-response data were analyzed as described in detail previously[41]. Unless indicated otherwise, differences between Y268S mutants and WT controls were assessed by the Mann–Whitney test or Fisher's exact test, in GraphPad Prism 9.

## Reporting summary

Further information on research design is available in the Nature Portfolio Reporting Summary linked to this article.

## Data availability

All data supporting the findings of this study are provided in the paper and its supplementary information. The minimum dataset necessary to interpret, verify and extend the research in this article are provided in the Supplementary Information/Source Data file. Whole genome sequence files for WT parent and Y268S mutant have been deposited in the NCBI Sequence Read Archive (BioProject ID: PRJNA913198; https://www.ncbi.nlm.nih.gov/sra). Source data are provided with this paper.

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

## Acknowledgements

This work was supported by Unitaid project LONGEVITY (2020-38-LONGEVITY); the Johns Hopkins Malaria Research Institute (JHMRI) and Bloomberg Philanthropies; and the National Institutes of Health (R01AI132359 to PS, R01AI1095453 to TS, and T32AI138953 to DS). RE was supported by a JHMRI postdoctoral fellowship. We thank Chris Kizito, Tassanee Thanakornsombut, and the JHMRI Insectary and Parasitology core facilities for their outstanding support; Dr. Sachie Kanatani for help with mosquito dissections and image development; the Clinical Pharmacology Drug Development Unit for weekly supply of healthy donor blood; Jodie Franklin and the JHMI Synthesis and Sequencing Facility for Illumina sequencing; and Mary Barry for meticulous laboratory maintenance. Kit Carson, Matt Ippolito and Kerry Stuart provided thoughtful help with methods. We thank Marcello Jacobs-Lorena for carefully reading and critiquing the manuscript. *P. falciparum*, Strain NF54 (Patient Line E), MRA-1000, contributed by Megan G. Dowler, was obtained through BEI Resources, NIAID, and NIH. We are grateful to staff at Yecuris for helpful advice and for generously replacing mice that unexpectedly died.

## Author contributions

Research was designed by R.P.B., A.O., S.T.P., T.A.S., P.S., D.S., and A.K.T. Experiments were conducted by R.P.B., V.A.B., M.A.C., A.G.D., R.E., A.E.J., A.M., G.M., E.N., K.R.-R., A.S., T.A.S., P.S., D.S., and A.K.T.. New analytical methods were devised by R.E. and S.T.P. Data were analyzed by R.P.B., V.A.B., A.G.D., R.E., A.E.J., E.N., S.T.P., K.R.-R., T.A.S., P.S., D.J.S., and A.K.T. All authors contributed to writing the manuscript.

## Competing interests

R.P.B., G.M., A.O., T.A.S., and A.K.T. are coinventors on PCT/GB2017/051746 (Atovaquone long-acting injectable formulation). A.O. is Director of Tandem Nano Ltd; received personal fees from Gilead and Assembly Biosciences; and is co-investigator on funding received by the University of Liverpool from ViiV Healthcare and Gilead Sciences. D. J. S. is Founder, Board Member, and stock/option owner of AliquantumRx; an expert legal consultant for Mabery Legal Firm; coinventor on USP 7,270,948 (Detection of malaria parasites by laser desorption mass spectrometry), USP9,568,471 (Malaria diagnosis in urine), USP 9,642,865 (New angiogenesis inhibitors), and PCT/US2015/046665 (Salts and polymorphs of cethromycin for the treatment of disease); and has received royalties from Binax Inc/D/B/A Inverness Medical for plasmids for HRP aldolase for malaria diagnostic kit. The remaining authors declare no competing interests.
