## [Peer Review File · Nature Communications]

Clinically relevant atovaquone-resistant human malaria parasites fail to transmit by mosquitoReviewers' Comments:

Reviewer #1:

Remarks to the Author:

In the manuscript titled "Transmissibility of clinically relevant atovaquone-resistant *Plasmodium falciparum* by anopheline mosquitoes," Balta and colleagues investigate the transmissibility of malaria parasites with clinically relevant, atovaquone-resistance conferring mutations in cytochrome b. Since atovaquone is being considered for use as a long-acting injectable antimalarial that confers causal prophylaxis, experiments on transmissibility of resistant parasites are vitally important due to concerns over drug resistance. Previous use of atovaquone to treat established erythrocytic infections in the field led to the emergence of resistance through mutations in cytochrome b, primarily at the Y268 codon. For this study, the authors generated isogenic lines of the NF54 strain of *P. falciparum* that differed only by a single mutation at this codon (Y268S), which is one of three amino acid substitutions associated with resistance in the field (the other two being Y268N/C). The authors then demonstrated experimentally that the Y268S mutation had fitness costs that reduced parasite growth in vitro (likely through fewer progeny within late schizonts) and, more importantly, eliminated parasite transmission by the mosquito. For the latter, the authors convincingly demonstrated that the Y268S mutation impacted nearly every sporogonic stage in the mosquito, including reduced exflagellation (suggesting an effect on microgametogenesis), a reduction in the number of ookinetes, near elimination of oocysts in the midgut along with reduced diameter of those present, and complete elimination of sporozoites in the salivary glands. The authors then demonstrated that mosquitoes infected with mutant parasites were incapable of transmission to mice engrafted with human liver cells and erythrocytes in stark contrast to wildtype parasites.

In addition to the mutation at Y268, which is the codon most often associated with high-level resistance to atovaquone, the authors also provided supplementary data on a different mutation that confers low-level resistance (L144S). In contrast to Y268S mutants, L144S mutants are capable of developing sporozoites that reach mosquito salivary glands, although reduced by 63% compared to wildtype. In the Discussion the authors suggest that fitness in the mosquito may be proportional to the degree of atovaquone resistance and that parasites with low-level resistance mutations would unlikely persist at plasma drug concentrations in atovaquone-treated hosts. In light of the experimental data, the authors conclude the manuscript by making a compelling case for a clinical trial of atovaquone implemented as a "chemical vaccine" in a malaria endemic setting.

The manuscript was very well written and the only typo I found was in the legend for Figure 3, where the word "mouse" should be omitted from the following: "(B) Indicated tissues were harvested from mice mouse bitten 7.5 d previously by mosquito...". My only other criticism is that the authors were only able to test a single mutation, but as they pointed out, the inability to obtain Y268N/C mutants in the lab may reflect a biological bias that operates for selection in vitro. Therefore, I recommend acceptance for publication.

Reviewer #2:

Remarks to the Author:

The paper describes trials of transmission by mosquitoes of human malaria parasites resistant to the mitochondrial electron transport chain inhibitor atovaquone. The data presented are wholly consistent with zero transmission of the resistance mutation Pf cytB Y268S, which is the only meaningful mechanism of resistance clinically found after decades of atovaquone use. This is important because it very strongly suggests that atovaquone could be used in various ways such as chemo-vaccination, prophylaxis, or as a partner drug to protect other antimalarial from succumbing to resistance, without creating a resistance tsunami. Considering that atovaquone is safe and cheap, this is good news.

I recommend several editorial changes. Next time, please use line numbers to make this exercise

easier for us all. I have supplied a PDF with my annotations, all of which require attention.

Page 1 – I find the title misleading. Indeed, on first reading I assumed that the mutation is transmissible. In the title, the use of the word ‘transmissibility’ refers to the topic rather than the conclusion. I recommend a change to “Clinically relevant atovaquone-resistance in human malaria parasites is not transmissible via mosquitoes”, which is more descriptive of the conclusion and avoids specialist jargon such as the Latin name of the parasite and the technically specific group of mosquito vectors, none of which the general reader needs to know to appreciate the headline finding.

Page 2 (and throughout). The paper does not follow established gene/protein nomenclature. The protein should be referred to as cytochrome *b* (with the ‘b’ in italics), and the gene should be referred to as *cytB*, with *cytB* all in italics. You could follow bacterial nomenclature and refer to the protein as CytB (no italics), but the mitochondrial literature favours cytochrome b (with the ‘b’ in italics).

Page 5 (and throughout)– both *in vivo* and *in vitro* are derived from Latin and should be italicised.

Page 5 – reference 20 is cited as reporting a Y268S mutation but also not reporting a Y268S mutation.

Line 8 – What does ‘no other significant genetic changes’ mean? This statement feels like a brush off. Put the differences in the supplementary data and let the readers judge if they are important. Whatever you do, don’t use ‘significant’, which is a very loaded term in science writing.

Page 8 & 10 – spell out numbers less than 10 unless used in conjunction with a unit of measure. i.e. eight not 8 in this instance on page 8 and five not 5 on page 10.

Page 13 – My English teacher used to beat us if we used two semi colons in one sentence unless they were used as separators for a colon designated list with sublist sections separated by commas.

Page 16 – For the second sentence, I believe you could include ‘oocyst diameter’ to the list of apparent fitness costs.

Page 16 – I strongly disagree with the use of ‘proportional’ to relate fitness in the mosquito to the degree of atovaquone resistance. Firstly, you only measured fitness in two mutants, Y268S and L44S, so you can’t claim proportionality from a slope with two points. Reword this to point out that the highest level of resistance has zero transmission whereas some transmission likely occurs in less resistant lines.

Page 3, 4, 9 & 10 (not numbered!) in the supplementary data – Please follow the conventions for restriction enzymes where the three-letter shorthand for the bacterium of origin is italicised but the Roman numeral for the particular enzyme from that source bacterium is not.

Page 5 in the supplementary data – I don’t understand supplementary Figure 3C. Please explain it more clearly.

Reviewer #3:

Remarks to the Author:

The paper from Victoria.A. Balta et al entitled “Transmissibility of clinically relevant atovaquone-resistant *Plasmodium falciparum* by anopheline mosquitoes” aims at showing that a resistant parasite strain to Atoquivone, - an anti-malarial which targets Cyt b-, although surviving to the treatment, is not able to be efficiently transmitted to *Anopheles* mosquitoes. This was both confirmed by mosquitos’

dissection at different time points after feeding of mosquitos with infected red blood cells in culture, and further assay in vivo on blood stage infection of humanized liver mice after biting by mosquitoes bearing mutant parasites.

These results were observed using a mutant parasite selected in vitro for its resistance to atovaquone based on a Y268S mutation in Cyt b, similar, although not identical mutation, to the one observed in patients.

These results therefore reinforced the remaining usefulness of this drug, which targets both the liver and the blood stages of the parasites, as transmission controller.

The manuscript is clear and well-written, and the results are convincing. Their scope is obviously relevant in a context of increasing resistance to anti-malarials and the risk of abandoning potentially still useful drugs. Atovaquone, that can be delivered as a long-acting injectable, could be a valuable tool to control parasite transmission, behaving as a "chemical vaccine". However, there are still some points to address.

Comments:

1- Were the mutant parasites cloned? Although the whole genome sequencing data seems to only identify Y268S mutation upon drug selection, can you formally exclude the involvement of other SNPs in the phenotype you obtained for the mutant parasites?

Have you ever tried to introduced the mutations found in the field (leading to Y268S,-N and -C codons) by CRISPR/Cas9? Although there are several copies of CytB, that should be feasible as they are homologous. These "artificial" mutants would be good controls for your demonstration.

2- To better evaluate the relevance of this work and the potential use of Atoquivone as a potential "chemical vaccine", it would be important to mention the frequency of mutations at this codon in the field in introduction and further discuss. Does this frequency vary between regions? Ethnicns? Is there environmental pressure for these mutations?

3- Regarding the production of gametocytes and transmission to mosquitoes, have you tried to inject immunodeficient mice (NSG for example) with RBCs, then infected RBCs, and make these mice bitten by mosquitoes? Indeed, the in vitro maturation of gametocytes could be somehow suboptimal and, while still sufficient for WT parasite transmission, it could be proven inefficient for less fit mutant parasite transmission and further maturation within Anopheles hosts.

4- One of the strengths of the manuscript is the in vivo experiments. However, I have some concerns about the detection of parasites in the infected humanized mice.

In Figure 3B, the electrophoresis gel looks faint, and the bands seems not optimally detected. The ladder for example cannot be properly visualized so that it may be difficult to conclude there is no signal for the mutant parasites.

In addition, it seems that parasites can be detected via Stevor amplification in both blood, liver and spleen but this appears not to be the case for CytB, although the signal obtained with CytB is higher than the one obtained with Stevor (if experiments are comparable). How can you explain this?

Quantification of parasite DNA products by qPCR (even if nested) using a standard curve established with genomic *P. falciparum* DNA would be useful to accurately compare both types of infection, in the different organs.

Have you by any chance kept some RNA from these experiments? qRT-PCR specific for 18S maybe more sensitive that the assays used.

5- Again, in Figure 7D, the authors provide a very interesting experiment mixing several ratios between WT and mutant parasites, which leads to the clear conclusion that WT outcompete mutant parasites in Anopheles host. Here again, it would be highly valuable to quantify the resultant mixture between mutant and WT parasites. Is there any mean to devise make a run on on the RFLP cut

products using fluorescent primers?

6- Figure 1B: the % effect on Y axis should be properly defined, at least in the legend.

7- Page 23, line 7: "Supplementary Fig. S1A,C" has to be changed into "Supplementary Fig. S2A,C".

8- It would be informative to put the standard deviation of the ookinete numbers in Supplementary Fig.4A or each to provide each individual value.

9- Idem for Supplementary Fig.5 to appreciate the variability of the sporozoite numbers between the mutant parasites.

RESPONSE TO REFEREES

We are grateful to the reviewers for their thoughtful, meticulous, and highly supportive remarks. As a consequence, the manuscript has been clarified and strengthened. Every comment is repeated verbatim below in *italics*, followed by our nonitalicized replies. Reviewer #2 also provided handwritten notes, each of which has been fully addressed, as described.

Reviewer #1

1.1 (line #684) *...in the legend for Figure 3, ... the word “mouse” should be omitted from the following “(B) Indicated tissues were harvested from mice mouse bitten 7.5 d previously by mosquito...”*

Thank you, this typo has been corrected.

1.2 *My only other criticism is that the authors were only able to test a single mutation, but as they pointed out, the inability to obtain Y268N/C mutants in the lab may reflect a biological bias that operates for selection in vitro.*

We share the reviewer’s disappointment on this matter. Not described in the manuscript, our efforts to generate Y268N or C mutants have involved several different methods, in vitro and in mice, some of which are quite unconventional, some of which are still ongoing, but none of which have worked to date.

Reviewer #2

2.1 *Next time, please use line numbers to make this exercise easier for us all.*

Line numbers have been added to the manuscript and are cited in these replies. However, we found that the manuscript's line numbers are changed in the process of uploading to the journal, even when a PDF version is supplied. This version of the response has been revised to match the uploaded manuscript file, in hopes they remain in register.

2.2 *I have supplied a PDF with my annotations, all of which require attention.*

This PDF was helpful in ensuring we attended to all concerns. Except for the three items below, all 56 annotations to the PDF of manuscript plus Supplementary Information have been fully addressed with the requested revisions, as now evident in track change.

Items

- (line #301 and 310) *Add hyphen between “clinically” and “relevant”.*
"Clinically relevant" appears 13 times in the text, starting with the title, but the suggestion to hyphenate was made only twice, starting in the Discussion. Compound adjectives are typically hyphenated, but not (as in this case) if they include an adverb. Accordingly we have not hyphenated "clinically relevant".
- (line #325) We are uncertain what the reviewer intended by the question mark inserted between "respiration" and the period that follows.
- Supplementary Figure S7 legend. *Do not italicise “Pie charts”.*
This term been retained here in italics, as it was italicised in the legends for manuscript Fig. 2A and Supplementary Figs. S6A and S9C. We ask the copy editor for journal style guidance on this matter.

2.3 (lines #1-2) *Page 1 – I find the title misleading. Indeed, on first reading I assumed that the mutation is transmissible. In the title, the use of the word ‘transmissibility’ refers to the topic rather than the conclusion. I recommend a change to “Clinically relevant atovaquone-resistance in human malaria parasites is not transmissible via mosquitoes”, which is more descriptive of the conclusion and avoids specialist jargon such as the Latin name of the parasite and the technically specific group of mosquito vectors, none of which the general reader needs to know to appreciate the headline finding.*

We thank the reviewer for this sensible recommendation, the title has accordingly been changed to: **Clinically relevant atovaquone-resistant human malaria parasites fail to transmit by mosquito**. This avoids specialist jargon and is not quite so absolute as the suggested title.

2.4 (line #32) *Page 2 (and throughout). The paper does not follow established gene/protein nomenclature. The protein should be referred to as cytochrome b (with the 'b' in italics), and the gene should be referred to as cytB, with cytB all in italics. You could follow bacterial nomenclature and refer to the protein as CytB (no italics), but the mitochondrial literature favours cytochrome b (with the 'b' in italics).*

Nomenclature is now standardized throughout, with **cytB** for the gene and **cytochrome b** for the protein.

2.5 (line #94) *Page 5 (and throughout)– both in vivo and in vitro are derived from Latin and should be italicised.*

These terms, and others derived from Latin, are now italicised throughout both manuscript and Supplementary Information.

2.6 (line #101) *Page 5 – reference 20 is cited as reporting a Y268S mutation but also not reporting a Y268S mutation.*

Reference 20 describes three different mutations in cytochrome *b*, one of which was Y268S. The other two were M133I and K272R.

2.7 (lines #139-143) *Line 8 – What does 'no other significant genetic changes' mean? This statement feels like a brush off. Put the differences in the supplementary data and let the readers judge if they are important. Whatever you do, don't use 'significant', which is a very loaded term in science writing.*

As now described in the text (and provided in Supplementary Data File 1) 3,991 variants were identified that had VarQual scores ≥ 30 . The VarQual (or Phred) score is defined as $-10 \log_{10} P$ (where P is the probability of an incorrect base call). VarQual score for the Y268S mutation is 49,191, so the probability of an incorrect base call is 10^{-4920} . The next highest score is 721. "Significant" no longer appears in this paragraph, since the numeric data are provided.

2.8 (lines #152, 181, 183, 190, 196) *Page 8 & 10 – spell out numbers less than 10 unless used in conjunction with a unit of measure. i.e. eight not 8 in this instance on page 8 and five not 5 on page 10.*

This has been done.

2.9 (lines #249-251) *Page 13 – My English teacher used to beat us if we used two semi colons in one sentence unless they were used as separators for a colon designated list with sublist sections separated by commas.*

In this sentence we use semicolons in coordination, a solution required by the length and complexity of the coordinates. Inasmuch as the coordinated phrases are long and/or contain internal punctuation, it would seem semicolons are justified by both the (draconian!) English teacher's allowance for sublist sections, and the latest edition of *The Cambridge Grammar of the English Language*.¹ The alternatives to serial semicolons – all commas or multiple sentences – would be much less effective. Since the sentence is grammatically correct we have opted not to make changes.

2.10 (lines #314-318) *Page 16 – For the second sentence, I believe you could include 'oocyst diameter' to the list of apparent fitness costs.*

We want readers to focus on the progressive fall in the number of mutant parasites as the lifecycle advances, rather than on the quality or size of various forms, so have not included oocyst diameter here.

2.11 (line #320-321) *Page 16 – I strongly disagree with the use of 'proportional' to relate fitness in the mosquito to the degree of atovaquone resistance. Firstly, you only measured fitness in two mutants, Y268S and*

L44S, so you can't claim proportionality from a slope with two points. Reword this to point out that the highest level of resistance has zero transmission whereas some transmission likely occurs in less resistant lines.

Both "proportional" and "proportionality" have been deleted. Our original proposal to the sponsor included studying a larger number of mutants so we could properly assess the relationship between degree of drug resistance and fitness cost, but that aspect was not supported. We suspect the actual relationship between resistance and fitness is sigmoidal (*i.e.*, a classic toxicity dose-response function) but agree we have insufficient data to support this. Interestingly, a plot of $\log_{10}EC_{50}$ (resistance) vs number of sporozoites/mosquito (fitness), yields a three point line for WT, L144S and Y268S, with $R^2 > 0.99$.

2.12 Page 3, 4, 9 & 10 (not numbered!) in the supplementary data – Please follow the conventions for restriction enzymes where the three-letter shorthand for the bacterium of origin is italicised but the Roman numeral for the particular enzyme from that source bacterium is not.

Apologies, we have corrected Word's page numbering glitch, and all pages in the supplementary information are now numbered. Restriction enzymes have been italicised as requested.

2.13 Page 5 in the supplementary data – I don't understand supplementary Figure 3C. Please explain it more clearly.

The legend for Supplementary Figure 3C has been augmented to address this question. We thought this information was interesting, but it is certainly not essential for the main findings of the study. If the revised version is still judged to be too obscure we would be happy to delete this panel.

Reviewer #3

3.1A Were the mutant parasites cloned? Although the whole genome sequencing data seems to only identify Y268S mutation upon drug selection, can you formally exclude the involvement of other SNPs in the phenotype you obtained for the mutant parasites?

Please see response to 2.7 above, lines #139-143 in the manuscript, and new Supplementary Data File 1. We did not clone the mutant before sequencing, but of nearly 4,000 sequence differences identified by VarQual, only Y268S had a substantial score. Score for the next closest variant was orders of magnitude less. We cannot categorically exclude the involvement of other SNPs, but our analysis identifies only Y268S as convincingly related to the phenotype.

3.1B Have you ever tried to introduced the mutations found in the field (leading to Y268S,-N and -C codons) by CRISPR/Cas9? Although there are several copies of CytB, that should be feasible as they are homologous. These "artificial" mutants would be good controls for your demonstration.

We have considered a number of methods to introduce mutations into the mitochondrial genome. It would be relatively easy to modify Cas9 for import into the mitochondrion, but there is no known mechanism to direct the guide RNA into this organelle. Furthermore, it is also not clear what genome maintenance mechanisms are used in the mitochondrion and whether traditional homologous recombination would work efficiently. The final conceptual problem is how to get the repair construct DNA containing the mutation into the mitochondrion.

3.2 To better evaluate the relevance of this work and the potential use of Atoquivone as a potential "chemical vaccine", it would be important to mention the frequency of mutations at this codon in the field in introduction and further discuss. Does this frequency vary between regions? Ethnicity? Is there environmental pressure for these mutations?

Most of these interesting questions do not have answers, in large part because (costly) atovaquone+proguanil is rarely used in endemic areas, so the pressure to establish resistance is not great. It is generally believed that atovaquone resistance is not commonplace in the field.² Certainly there are no reports of established resistance,

or of the global genetic sweeps we have seen for resistance to chloroquine or the antifolates. The relevant literature on atovaquone resistance comprises case reports of patients who failed atovaquone±proguanil treatment of symptomatic erythrocytic malaria. Of these, 23 primary publications include 24 examples of Y268S, 14 of Y268C, and 2 of Y268N. Although the largest number of reported mutants are Y268S, because these reports are so anecdotal we are reluctant to claim a relationship between them and our lab findings. In only five instances were parasites cultured so as to obtain EC₅₀ values; the remaining reports featured sequence changes only. Two reports surveyed limited regions of Africa, and most information on this subject is likely not readily available (i.e., unpublished periodic within-country surveys by Ministries of Health). In terms of possible environmental pressure for these mutations, atovaquone alone (Mepron) is used to prophylax and treat immunocompromised people with *Pneumocystis* pneumonia or toxoplasmosis, so patients co-infected with HIV and malaria could pose an environmental pressure.

3.3 Regarding the production of gametocytes and transmission to mosquitoes, have you tried to inject immunodeficient mice (NSG for example) with RBCs, then infected RBCs, and make these mice bitten by mosquitoes? Indeed, the in vitro maturation of gametocytes could be somehow suboptimal and, while still sufficient for WT parasite transmission, it could be proven inefficient for less fit mutant parasite transmission and further maturation within Anopheles hosts.

In an effort to select atovaquone-resistant *P. falciparum* *in vivo*, we have tried long and hard, so far without success, to establish a sustained infection in NSG mice with wild type asexual erythrocytic parasites that we know retain the ability to generate functional gametocytes *in vitro*. Reports of *P. falciparum* gametocytes obtained in a murine model are rare and we know of none that have demonstrated transmissibility of these parasites to mosquitoes.

3.4A One of the strengths of the manuscript is the in vivo experiments. However, I have some concerns about the detection of parasites in the infected humanized mice. In Figure 3B, the electrophoresis gel looks faint, and the bands seems not optimally detected. The ladder for example cannot be properly visualized so that it may be difficult to conclude there is no signal for the mutant parasites.

Ten-fold longer exposures of the entire gel fields have now been added to the supplementary information (Fig. S10), so interested readers can see that the described findings are unambiguous. We did not include these images in the main manuscript for sake of space and because in the absence of cognate template DNA the *Stevor* reaction (with its complex primer mix) generates an assortment of spurious bands, which take some effort for the reader to evaluate.

3.4B In addition, it seems that parasites can be detected via Stevor amplification in both blood, liver and spleen but this appears not to be the case for CytB, although the signal obtained with CytB is higher than the one obtained with Stevor (if experiments are comparable). How can you explain this?

With WT samples, bands are detected for all three tissues (blood, liver, spleen) by both *Stevor* (Panel B) and *cytB* (Panel C). (See also new Fig. S10.) There are, however, differences in relative band intensities between the *Stevor* and *cytB* amplifications, because each assay was conducted with tissues harvested from a different mouse, one from each replicate experiment. This is noted in the legend and was done on purpose so as to assure readers that both replicate experiments fully support the conclusions.

3.4C Quantification of parasite DNA products by qPCR (even if nested) using a standard curve established with genomic P. falciparum DNA would be useful to accurately compare both types of infection, in the different organs.

Relative amounts of DNA obtained by qPCR would indeed be interesting to know, but to obtain these data in a rigorous manner we would have to assess the impact of carryover matrix contaminants that affect amplification (which we found is different among these three tissues), and also somehow normalize tissue samples to one

another (but on what basis? Total DNA, tissue weight, other?). We therefore thought it most reasonable to obtain the indicated lower limits of detection (line #464) and apply them for a categorical outcome.

3.4D *Have you by any chance kept some RNA from these experiments? qRT-PCR specific for 18S maybe more sensitive than the assays used.*

We unfortunately isolated only DNA from mouse tissues. However, it seems unlikely that we failed to detect mutant parasites in every attempt that was made. For each mouse we conducted a total of seven assays, involving three tissues and three sensitive and independent methods (*in vitro* culture of total blood volume, and nested PCR for nuclear DNA-encoded *Stevor* or mitochondrial DNA-encoded *cytB*). This was done in two biological replicate experiments.

3.5 *Again, in Figure 7D, the authors provide a very interesting experiment mixing several ratios between WT and mutant parasites, which leads to the clear conclusion that WT outcompete mutant parasites in Anopheles host. Here again, it would be highly valuable to quantify the resultant mixture between mutant and WT parasites. Is there any mean to devise make a run on on the RFLP cut products using fluorescent primers?*

Yes, this experiment turned out to be more interesting than we had anticipated. The RFLP results evidence no mutant DNA in either the midguts or salivary glands. As described in the legend for Supplementary Figure 2D, these assays can detect an allelic population as low as 2% (*NsiI*) or 10% (*PstI*). If present, mutant DNA must therefore be at less than 2%. The legend to Supplementary Figure 7D has been amended to direct the reader's attention to the methods and sensitivities for the RFLP methods that are detailed in the legend for Supplementary Figure 2D.

3.6 *Figure 1B: the % effect on Y axis should be properly defined, at least in the legend.*

The legend has been amended to indicate that Y-axis % Effect was assessed by changes in [³H]hypoxanthine incorporation.

3.7 *Page 23, line 7: "Supplementary Fig. S1A,C" has to be changed into "Supplementary Fig. S2A,C".*

Thank you for finding this error, correction has been made in line #458.

3.8 *It would be informative to put the standard deviation of the ookinete numbers in Supplementary Fig.4A or each to provide each individual value.*

These results are from one experiment so we have no standard deviations; however, the figure has been revised to depict the individual values.

3.9 *Idem for Supplementary Fig.5 to appreciate the variability of the sporozoite numbers between the mutant parasites.*

These data were obtained in a single experiment, hence standard deviations are not available.

REFERENCES CITED

1. Huddleston R, Pullum GK (2016) Punctuation 3.2.1 Coordination, syndetic or subclausal. Use of the semi-colon in coordination. Chapter 20 In, *The Cambridge Grammar of the English Language*, p 1740. Cambridge University Press, Berforts Information Press Ltd, UK

2. World Health Organization. *World Malaria Report 2022* (World Health Organization, 2022)

Reviewers' Comments:

Reviewer #3:

Remarks to the Author:

The rebuttal is satisfying and the new version submitted by the authors are much improved in several ways, including the title change asked by another reviewer, which is indeed better adapted to the data. All the points raised were correctly addressed and this interesting paper is now suitable for publication in Nat. Com.

RESPONSE TO REFEREES

Reviewer comment is repeated verbatim below in *italics*, followed by our nonitalicized reply.

Reviewer #1

No comments were sent

Reviewer #2

No comments were sent

Reviewer #3

The rebuttal is satisfying and the new version submitted by the authors are much improved in several ways, including the title change asked by another reviewer, which is indeed better adapted to the data. All the points raised were correctly addressed and this interesting paper is now suitable for publication in Nat. Com.

We thank the reviewer for these supportive comments.